# The Impact of Dynamic Accounting Information System on Organizational Resilience: The Mediating Role of Business Processes Capabilities

**Ahmed Saleh Al-Matari** [1,2,*], **Rozita Amiruddin** [1], **Khairul Azman Aziz** [1] and **Mohammed A. Al-Sharafi** [3,*]

[1] School of Accounting, Faculty of Economics and Management, Universiti Kebangsaan Malaysia (UKM), Bangi 43600, Selangor, Malaysia; rozita@ukm.edu.my (R.A.); khairul.aziz@ukm.edu.my (K.A.A.)

[2] Faculty of Commerce and Economics, Sana'a University, Sana'a 15542, Yemen

[3] Department of Information Systems, Azman Hashim International Business School, Universiti Teknologi Malaysia, Skudai 81310, Johor, Malaysia

[*] Correspondence: ahuniv2005@hotmail.com (A.S.A.-M.); alsharafi@ieee.org (M.A.A.-S.)

**Abstract:** For decades, one of the main concerns of both practitioners and academics has been the business value of dynamic accounting information systems (DAIS). A number of studies have demonstrated the positive effects of information systems capability on overall organizational performance, but our understanding of the business processes capabilities through which such gains are achieved remains limited due to a lack of focus on the turbulent business environment. As a result, the research on information systems continues to debate such a connection. The role of business process capabilities in modulating the link between dynamic AIS capability and organizational resilience was investigated in this study. Our results show that, while firm-wide dynamic AIS capability has characteristics of flexible AIS, complement BI system, and AIS-related human resource competency, the impact on organizational resilience is positively affected by mediation of business process capabilities based on 144 matched questionnaires selected from large companies from various sectors listed on the Bursa Malaysia. Our results also suggest that dynamic AIS capability has an impact on organizational resilience. According to the Resource-Based Theory (RBT) and dynamic capabilities view (DCV) viewpoints, there is a link between dynamic AIS and business process capacity to improve organizational resilience. The findings strongly support the claim that an organization's dynamic AIS capabilities—both flexible AIS, complementary business intelligence (BI) system, and AIS-related human resource competency—can help an organization improve its resilience. This research's practical and theoretical ramifications as well as its limitations are examined.

**Keywords:** accounting information system; dynamic accounting information system; organizational resilience; business processes capabilities

## 1. Introduction

Crises are unavoidable in today's turbulent business climate, but there are significant variances in how organizations deal with problems or disruptions, both internal and external, that could lead to a crisis [1]. According to the United Nations trade and development organization, the economic uncertainty generated by the coronavirus outbreak is expected to cost the world economy USD 1 trillion by 2020. These crises have become one of the most important predictors of volatility impacting an organization's performance as well as one of the leading causes of insolvency or bankruptcy [2]. During all crises, organizational activities are disrupted by a multitude of strategic, environmental, financial, or political causes. Thus, while interruptions and other crises are unavoidable, there are significant variances in how organizations deal with them. Information systems (IS) academics have recently advocated that organizations should enhance their IS capability to obtain a competitive advantage and face the dynamic and turbulent character of the organization environment

based on the resource-based view (RBV) and dynamic capabilities theory [3–5]. The concept of dynamic accounting information system capability, at its core, emphasizes the necessity of integrating, creating, and reconfiguring IT-based resources alongside and leveraging the value of other resources and capabilities to reorganize accounting activities and processes quickly. Dynamic capabilities, according to empirical research, contribute to organizational performance [6–8].

In today's fast-paced and chaotic business climate, having a dynamic AIS capability is critical [9]. This necessitates organizations focusing more on day-to-day operational demands at the price of long-term resilience and digital business innovation [10]. As a result, AIS and IT have evolved into metaphors for many tools and approaches that organizations might use to tackle the challenges of these environments. As a result, many investments in information systems (ISs) are undertaken in the belief that certain technologies, such as accounting information systems (AIS), would pay off, proving essential to an organization's competitive survival [11]. These crises affect organizations' continuity and threaten their survival and competitiveness, increasing their chances of collapsing [12]. They are also a major challenge for management and for information systems. Thus, the lack of resilience among organizations may be a result of the lack of dynamic AIS capability among the organizations [13–15]. This subject is somewhat unexplored in the context of major corporations and the importance of dynamic AIS capability. As a result, the impact of dynamic AIS capacity on organizational resilience is investigated in this research [16,17]. This study focuses on enhancing the AIS's dynamic potential through the collaboration of a flexible AIS, a complementing BI system, and AIS-related human competencies.

Some academics suggested that business processes could be crucial determinants in linking IT-resources-based capabilities and organizational success, based on RBV theory, [18,19]. An important aspect of internal and external business processes is that of business process capabilities, i.e., outside-in, inside-out, and spanning [20]. These terms refer to the organization's ability to create added value in a unique way to utilize resources [21].

A significant amount of IS research has focused on either IT investment or IT capability, which is described as Information Technology and associated hardware, software, and services [22]. However, AIS is not a component of IT only, as it also deals with procedures and structures, internal control, and processes and people in order to manage the information effectively [23]. The actual usage of such systems is important, and it is the missing link between IT/IS and firm performance [24–26]. Two of the most critical areas indicating the effective usage of AIS and its impact on firm performance are the decision-making processes and business processes capabilities [27]. Therefore, to explore the strategic role of dynamic AIS capability, this study investigates the relationship between dynamic AIS capability and business-process capabilities. In addition, a number of studies have demonstrated the positive effects of information systems capability on overall organizational performance [28–31], but our understanding of the business processes capabilities through which such gains are achieved remains limited due to a lack of focus on the turbulent business environment.

Based on the resource-based view (RBV) and dynamic capabilities views, we propose that dynamic AIS capability can be constructed through flexible AIS, business intelligence (BI), and complementary and associated competencies (DCV). As a result, BI systems, which are one of the dynamic AIS components, are seen as excellent potential for businesses to decrease risks and increase profits [32] and give the analytical capabilities required to process data in the required manner [9]. Organizations may rely on a flexible AIS to keep their AIS applications up to date in real-time to adapt to the rapidly changing in real-world activity [9]. A dynamic capability requires a huge amount of learning [33,34]. As a consequence, IS-related human resource capability can also facilitate this learning in a dynamic AIS capability [9] and maintain the dynamic nature of it [27]. As a result, dynamic AIS capacity is seen as "a particular sort of resource, namely an organizationally

embedded non-transferable organization-specific resource that necessitates an examination of its impact on business processes" in this study.

For this study, the dynamic AIS capacity is defined as "a unique sort of resource, specifically an organizationally embedded non-transferable organization-specific resource that necessitates an examination of its impact on business process capabilities (BPC)". This study also looks at the influence of dynamic AIS capabilities on OR by identifying the research gap in AIS business value and its impact on OR as mediated by its impact on BPC. This study also looks into how dynamic AIS capability affects OR by identifying a research gap in AIS business value and its impact on OR as mediated by BPC.

## 2. Literature Review

### 2.1. Dynamic AIS Capability

Dynamic capabilities play a key important role in managing uncertainty [35]. Organizational dynamic capabilities are defined as an organization's overall ability to integrate, build, and reconfigure internal and external competencies in order to respond to quickly changing circumstances [34]. Hence, the current dynamic and turbulent business environment demand a dynamic AIS capability to survive [9]. For this study, a dynamic AIS capability implies the organizational ability to integrate, build, and reconfigure its competencies to swiftly reorganize accounting processes and activities [9]. Hence, a dynamic AIS capability can be developed from the positive interaction between AIS and related organization competencies [9].

Despite the fact that various AIS capabilities have been recorded in the literature, this research will concentrate on dynamic AIS capability in core organizational functions. Processing transactions, accounting information display for decision making, and control environment management are examples of such areas, with the flexible AIS, complementary BI system, and AIS-related human resource competency as engaged skills. The capacity of dynamic AIS was chosen for two reasons: To begin with, emphasizing the relevance of dynamic skills would be consistent with earlier IS strategy studies [9]; and second, despite their strategic importance, dynamic AIS capability has not been regarded in the OR of organizations' performance research.

In dynamic or uncertain times, organizations fail due to lack of resilience [36] and lack of dynamic AIS [9]. Researchers such as [13–15,37] suggested that lack of dynamic AIS capabilities as a factor to lack of resilience among organizations. Dynamic AIS capability is a critical resource that is expected to facilitate the identification of threats and opportunities; these capabilities transform existing strategies, structures, and innovations to take advantage of opportunities or resists threats, thereby achieving organizational resilience. This role of dynamic AIS capability, however, is relatively unexplored in the context of large organizations. Hence, this study investigates the impact of dynamic AIS capability on organizational resilience mediated by business process capabilities.

For the purpose of this study, a dynamic AIS competence is the result of a collaboration of three skills: (1) a flexible AIS, (2) a supplementary BI system, and (3) AIS-related human resource competency. A flexible AIS refers to the extent to which the technological elements of an organization's AIS are assessed on a routine basis as well as the ease at which the AIS could well be updated and the financial resources allocated to do so [9]. Although the BI system is an important dynamic aspect of AIS that allows firms to organize their data, produce knowledge from it, and implement strategies to preserve or gain a competitive advantage, it is not the only dynamic element [38]. The ability to leverage the AIS in unique ways that link to accounting professionals' technical IT skills is described as an AIS-related human resource capability [9,39].

In such uncertain and unstable times, organizations need to develop a resilience capability that enables them to deal effectively with unforeseen events, recover from crises and even promote future success, and take advantage of events that could threaten the survival of an organization [40,41]. With organizations facing business continuity, cash flow problems, supply chain challenges, business operations interruption, etc., it is now more

important than ever that they have a solid basis for their business systems. A modern suite of IT applications with AIS can provide a comprehensive, agile, secure, and integrated solution in a range of functions, such as finance and accounting, procurement, human resources, and sales, and by adopting this system, its business can continue regardless of interruption. Dynamic technology-based AIS could contribute to building resilient organizations.

Considering IS and strategic planning theoretical and operational perspectives, AIS is a vital factor in reducing uncertainty and increasing confidence in managing crises and minimizing their implications. It also provides prospects of eliminating the crises before their occurrence [42]. As such, the affected companies need to adapt rapidly and cope with the external and internal environments as well as be able to anticipate, prepare for, respond to, and adapt to the unexpected changes and disruptions to improve their chances of survival [43]. This, in turn, requires investing in developing and improving their technological capabilities [44,45]. Organizations should also consider the creative use of IS, such as dynamic AIS [27]. Because IT capabilities are scarce, valuable, and tough to copy, they have also shown to provide such a competitive edge [46].

Organizational growth and survival are dependent on their ability to ensure effective use of such data volume from various sources to achieve their operational and strategic goals [47–49]. Through investment in technological capabilities, creating synergy between flexible AIS, BI system, and AIS-related human resource competency and organizational competencies, they would improve their capacities to improve accounting processes, financial and non-financial reporting, as well as to make clear decisions in a timely manner, respond quickly to external and internal threats, identify opportunities, manage the control environment, and address ambiguity in different situations that, in turn, improve organizational resilience.

*2.2. Business Process Capabilities and Accounting Information System*

Business process capabilities, in this paper, refer to the ability of an organization to create market value by utilizing resources in a unique way [50]. Scholars have proposed a three-denomination typology for BPC classification: outside-in capability, inside-out capability, and spanning capability [20,51].

An organization's ability to anticipate market demands, filter out competitors, build a long-term strategic connection with external stakeholders, and respond to quick market changes is known as outside-in capability [51]. As per Wade and Hulland [52], the outside-in capability is the organizational ability to focus and emphasize the threats and opportunities in the external environment for onward corroboration with the internal processes. According to Banker et al. [53], an organization's ability to pursue operational excellence and efficiency through internal processes is known as inside-out capabilities. The merging of outside-in and inside-out organizational capability is referred to as spanning capability. The ability of an organization to handle information flow across functional areas of the supply chain (buying, order processing, strategy development, and information dissemination) is referred to as spanning capabilities [54]. Spanning capabilities rely on internal and external analyses to assist organizations in utilizing valuable strengths, exploiting market opportunities, avoiding potential weaknesses, and neutralizing external threats.

Both RBT and DCV have emphasized the necessity of dynamic capability-building procedures in attaining organizational competitive advantage [34,55]. Similarly, organizational performance can be affected by IS capability in all situations, including during times of crisis, via the mediating role of capabilities such as BPC [56]. In addition, organizational performance can be enhanced by an IT-based system, such as AIS capability, with the organization's ability to optimize BP and improve BP management [57]. Despite being a valuable resource in improving organizational performance, AIS on its own might not be able to help in sustaining that performance [58]. This is consistent with the RBT essence, which indicates that the effects of AIS as a valuable resource might still be dependent on intangible factors such as BPC [59]. However, better performance can be generated through the synergism of the corresponding tangible and intangible resources [60].

According to Liang et al. [59], organizational capabilities, such as BPs capabilities, are the most well-regarded mediators. To develop a causal relationship between BPC and OR, dynamic AIS systems enhance important interdependency between inputs, processes, and outcomes. Powell and Dent-Micallef [61] provided evidence that IT-based systems such as AIS can enhance organizational performance when complemented with other BPs. Based on this, BP improvements capabilities are expected to function as a mediator in the relationship between AIS capabilities and OR. Furthermore, the BI system's AIS capabilities must be used to improve an organization's overall performance in order to obtain accurate, relevant, and valuable information for managing daily activities and strategic objectives in a rapidly changing business environment [62].

## 3. Model Development and Hypotheses

### 3.1. Dynamic AIS Capability (DAISC) and Organizational Resilience (OR)

Based on the RBT perspective, AIS resources that are valuable and inimitable can be rent-yielding [63]. There are several IT-based IS in the modern era, such as accounting software programs, transaction process systems, control systems, BI, and crisis management software. However, access to these resources by itself cannot guarantee the resilience and sustainability of a company [64]. Therefore, the synergy between various organizational resources is a key aspect of OR [65]. In general, firms' IT systems serve as a vital antecedent for generating more competitive actions through channel integration [66]. In the current study, complementarities between a flexible AIS, complementary BI system, and AIS-related human competency have led to sustainable performance benefits.

Previous studies do support the impact of dynamic AIS, an IT-based system technical innovation, on OR. Oh and Teo [66] discovered that IT capacity has a greater impact on the organization's resilience. Furthermore, several authors [9,27,67] pointed out the significance of IT/AIS capabilities in enhancing organizational performance or resilience. As a result, we propose that dynamic AIS capacity distinguishes organization performance for survival, allowing the company to develop valuable capabilities and advantages in the ever-changing big data environment. As a consequence, we propose the following as our major hypothesis:

**Hypothesis 1 (H1).** *Dynamic accounting information system capability has a positive impact on organizational resilience.*

AIS-IT infrastructure flexibility can influence competitive advantage via OR. A flexible AIS application also positively supports dynamic AIS capability by providing a ready platform for accessing the appropriate data and establishing a network communication system for easy communication with other systems. An AIS, for instance, should have the capability to receive and process information from service delivery channels or new sales. Managers' expectations for a wider range of standard and ad hoc accounting information with changing levels of detail should be addressed by a dynamic AIS capability. All the organization sections should adapt to business needs changes and approaches by adapting and integrating the IT infrastructure. Thus, the infrastructure is an integral aspect of the IS capabilities for reaching every point and covering the range of the organizational boundaries [68].

Furthermore, online business models (such as business-to-business, business-to-consumer, and, more recently, consumer-to-consumer) necessitate a continuous flow and sharing of client data across all channels in order to assure consistency across databases. [66]. As such, a flexible AIS could be facilitated for the adaptability and reconfiguration of organizational processes [69]. This boosts an organization's ability to innovate and quickly adapt to a changing environment. As a consequence, organizations with more IT infrastructure flexibility are predicted to have better levels of organizational resilience. As a result, we propose the following sub-hypothesis:

**Hypothesis 1.1 (H1.1).** *A flexible accounting information system has a positive impact on organizational resilience.*

In order to adapt to today's fast-changing and dynamic environment, businesses must be agile [70]; thus, accounting information should be provided to decision makers on an ad hoc basis at various levels of detail [23]. A generic accounting system that lacks the ability to "slice and dice" accounting data also lacks the capability to meet these needs [71]. To overcome the limitations of generic AIS, enterprises should implement a BI system to supplement the AIS's data analytic capabilities. As a result, BI at a DAIS capacity plays a critical role in strengthening OR and its associated capabilities [72]. Furthermore, these capabilities are considered assets that facilitate effective organization adoption of BI [73]. Currently, BI systems are increasingly adopted to provide better analytical capabilities from the earlier-installed AIS systems used to manage huge organization-related information [74].

Organizations should be able to respond fast to unforeseen developments in terms of adaptability capabilities, according to the concept of OR. As just a result, organizations must use the information generated by BI as a valuable element to successfully deal with these challenges [75]. Because BI consists of internal and external information, business market information, and analysis [76], it will help managers make decisions quickly and improve their resilience. It also provides strategic managers with the necessary information about the current direction and possible changes in the future [77]. Strategic decisions, such as strategic resilience, are frequently unique, needing creative decision-making approaches and the use of data sources, decision support systems, and techniques [70]. Assistance for managing risk via a business intelligence system is essential, particularly for organizations operating in high-risk contexts [70]. Based on the foregoing considerations, we examined the relationship between BI systems and organizational resilience by considering the information supplied and the BI system's potential to innovate in order to solve problems and respond to challenges. As a result, we offer the sub-hypothesis below:

**Hypothesis 1.2 (H1.2).** *A complementary BI system has a positive impact on organizational resilience.*

Professional capabilities are critical to the development of dynamic capacities and the basic redevelopment of the useable resource base [78]. Organizational staff can only meaningfully connect purpose with the circumstance when they are aware of the organization's goal (i.e., OR) [79]. Therefore, human resource competency contributes significantly to dynamic AIS capability development [27]. Professional staff can also coordinate real-time responses to unexpected events or indirectly coordinate such responses via a common reference point (such as dynamic AIS capability) for new actions. This allows strategic coherence even with unplanned innovative actions [80].

Professionals can also assist organizations in developing greater capacities to examine the environment on the boundaries [81]. The goal of doing this is to enhance sensing of ongoing changes. This will act as the mechanism for acquiring increased organizational effectiveness [82]. As stated earlier in previous literature [63,83], skills and specificity are the two basic characteristics of AIS-related human resources. As these AIS-related human resource characteristics enable quick and easy communication between AIS staff and other business staff, we propose the following sub-hypothesis:

**Hypothesis 1.3 (H1.3).** *AIS-related human resources competency has a positive impact on organizational resilience.*

### 3.2. Business Process Capabilities Mediating the Impact of Dynamic AIS Capability on Organizational Resilience (OR)

We recognize that several characteristics of the RBV provide IS researchers with unique and beneficial benefits. To begin with, the RBT simplifies the defining of AIS resources by providing a standardized set of resource properties [52]. The RBT theory assumes that valuable, rare, non-imitable, non-substitutable, and organized internal resources and capabilities play a significant role in helping firms enhance their performance and create value and competitive advantage [84]. These resources include AIS resources consisting of (1) tangible assets, such as the AIS infrastructure and human resources (i.e., AIS-related human and AIS infrastructure), and (2) intangible assets, such as AIS-related human skills, competencies, and business processes [27].

According to RBV, an organization's resources are described as all assets, capabilities, organizational processes, organizational traits, information, and knowledge under its control that allow it to increase its efficiency and effectiveness [84]. RBV assumed that ownership and control of strategic assets (such as the dynamic capacity of AIS) determine which organizations will gain higher profits and enjoy a position of competitive advantage over the others. The RBV sets a new link between AIS resources and long-term competitive advantage, with a well-defined dependent variable to back it up, making it an effective tool for determining the strategic value of AIS resources [52]. Therefore, organizations may not easily replicate their rival organizations' strategies due to this assumption, as they cannot replicate their resources easily [84]. However, these differences in resources may persist, allowing the benefits from heterogeneous resources to persist over time [85]. As a result, an organization can be thought of as a collection of resources, talents, or processes that produce value that is difficult to duplicate due to unique characteristics [86].

Organizational competencies or capabilities are resources in themselves or in combination with other resources across organizational divisions [87]. Therefore, suggesting the dynamic capability of the AIS depends on the synergies between three competencies: the flexible AIS, the complementary BI system, and AIS-related human resource competency. As mentioned above, the resources are the set of available factors that are controlled by the organization. While the term capabilities refers to the ability to combine resources and implement processes to achieve the intended result of having knowledge, tangible or intangible aspects, and extraordinary business processes as well as complicated relationships between existing resources [88].

Consistent with previous studies, IS capabilities indirectly encourage better business-process performance by leveraging the other organizational capabilities and resources [89–91]. The essential indicators of business process capabilities are the rates of order fulfilment, operational efficiency, customer intimacy, and satisfying consumer expectations [27]. In addition, IT-based AIS creates business value at the operational level via its three separate but complementary impacts on business activities: (1) automation effects, referring to IT-based capability to extract value from the efficiency perspective via substitution of capital asset with labor and via cost reduction; (2) information effects, referring to IT capability in information storage, processing, and communication; and (3) transformational effects, referring to IT capability in encouraging process transformation and innovation [92].

Various studies have focused on the impact of dynamic AIS capabilities on business processes. Accordingly, various IS-related studies have shown that successful investment in IT infrastructure can positively revolutionize business processes and improve performance [27]. These positive changes can be experienced as direct and indirect effects between a dynamic AIS capability (i.e., flexible AIS, BI, and AIS human skills) and business process capabilities. As per Elbashir et al. [74], AIS assists organizations in creating business value, as they directly affect business processes. Such impact could be observed as improved BP efficiency and effectiveness [57,91]. Gu and Jung [11] highlighted the six areas of value chain that IS capabilities improve and that support core organizational business activities: (1) sales and marketing support, (2) supplier relations, (3) production

and operations, (4) product and service enhancement, (5) process planning and support, and (6) customer relations.

According to Liang et al. [59], organizational capabilities such as BPs capabilities are the most well-regarded mediators. Dynamic AIS systems promote critical interdependency between inputs, processes and outcomes to create a causal link between BPC and OR. Powell and Dent-Micallef [61] provided evidence that IT-based systems such as AIS can enhance organizational performance when complemented with other BPs. Based on this, BP improvements capabilities are expected to function as a mediator in the relationship between AIS capabilities and OR. In addition, the AIS capabilities enabled by the BI system must be leveraged to enhance an organization's performance in order to receive accurate, relevant, and insightful information to manage daily processes and strategic objectives in a dynamically changing business environment [62].

Several studies [91,93,94] revealed that the impact of IS capabilities could manifest as improved efficiency and effectiveness of business processes, as they directly affect business processes capabilities. Furthermore, Gu and Jung [11] identified six value chain areas where IS capabilities could improve and support core organizational business activities: (1) sales and marketing support, (2) production and operations, (3) process planning and support, (4) supplier relations, (5) product and service enhancement, and (6) customer relations. Hence, it is reasonable to assume that BPs play a mediating role in the positive direct effect of dynamic AIS capability on an organization's OR.

Business process capabilities, according to the RBT, are the most significant resources that enable organizations to attain a competitive edge. [20]. Through the role of mediation of those other assets or capabilities, AIS's dynamic capability can influence an organization's resilience. As a strategic capability, business process capabilities are contingent on an organization's ability to implement and exploit AIS resources [39]. The combination of these arguments suggests that the capabilities of the business process mediate the relationship between the dynamic capability of AIS and the organizational resilience of an organization. As a consequence, as demonstrated in the research model, we propose the following hypothesis:

**Hypothesis 2 (H2).** *Business process capabilities mediate the relationship between the dynamic accounting information system capability and organizational resilience.*

## 4. Methodology

### 4.1. Research Context and Data Collection

This study was undertaken deductively to analyze the relationship between facts and observable occurrences, and it is based on a positivist philosophy. The impact of dynamic AIS capacity on organizational resilience was investigated using a quantitative method. This study selected large companies in Malaysia as a study sample due to the argument that medium-to-large organizations are more able to provide AIS and more likely to have resources to ensure the organization's resilience [95]. Indeed, large organizations are a suitable sample for this study because they have required AIS capability and financial resources to conduct the synergies that constitute dynamic AIS capability. For companies listed on the stock market, the fact that those organizations are approved to be publicly traded shows their financial standing and their ability to compete with international organizations. Thus, it should be necessary for these organizations to build dynamic AIS and subsequently have considerable dynamic AIS experience and possibly have reached stability in their process, which could improve the organization's resilience. Hence, large organizations in Malaysia were chosen because (1) there are more available data on these organizations, and (2) in general, larger organizations have more intensive information and practice dynamic AIS more actively [96] and provide the broadest possible use of information-technology-based AIS in an enterprise resource planning environment since it covers all key business functions [97]. The initial sampling framework was drawn from companies listed on Bursa

Malaysia (https://www.bursamalaysia.com/ (accessed on 15 August 2020). This sample frame represents large Malaysian organizations in different industries.

The targeted respondents are chief executive officer, chief finance officer, director of information, director of IMIS, senior system analysts, director of database administration, IT managers, and senior accountants. The selection was based on the assumption that these individuals are generally involved in resource repurposing and innovation in the organization and thus are likely to have a better understanding of the organization's resource endowments, processes, and capabilities as well as their impact on the organization's innovation performance, suggesting a high level of accuracy in responses [98].

Survey questionnaires were distributed to 984 listed companies in Malaysia during the 10-month data collection period from December 2019 to September 2020. The duration of data collection was longer, as difficulties arose due to the COVID-19 pandemic. A total of 162 completed questionnaires were returned, giving a response rate of 16.5 percent. Out of 162 questionnaires received, 13 questionnaires were excluded because the respondent did not fulfill the requirement. The final sample consisted of 144 matched surveys after excluding unmatched and/or missing cases.

The respondents are mainly CFO (45.8%) and others (29.2% includes staff accountant, senior accountant, senior auditor), as shown in Table 1. The remaining respondents were the CIO, director of MIS, senior system analyst, and database administration director. This was not unexpected, as the study included those who are likely to use AIS systems to perform their tasks. Accordingly, the highest number of respondents comprised CFOs and staff of accountants who were more often using the dynamic AIS systems.

**Table 1.** Respondents' job position.

| | Frequency | Percent | Valid Percent | Cumulative Percent |
|---|---|---|---|---|
| CFO | 66 | 45.8 | 45.8 | 45.8 |
| CIO | 13 | 9.0 | 9.0 | 54.9 |
| Director of MIS | 13 | 9.0 | 9.0 | 63.9 |
| Senior System Analyst | 7 | 4.9 | 4.9 | 68.8 |
| Database Administration Director | 3 | 2.1 | 2.1 | 70.8 |
| Other | 42 | 29.2 | 29.2 | 100.0 |
| Total | 144 | 100.0 | 100.0 | |

*4.2. Measurement Items*

The questionnaire instrument was developed based on literature reviews. To measure each construct, the questionnaire items were adopted from previous empirical research. Using previously validated and published items increased and guaranteed the content validity and reliability of the items used for this study's constructs. The steps involved in developing the questionnaire are as follows: (1) reviewing literature in research to identify previously validated and published measures, (2) developing the initial draft of the questionnaire, (3) reviewing the questionnaire draft by practitioners and academics at pre-test stage, (4) conducting the pilot test, and (5) improving the questionnaire before sending for actual collection. On a five-point Likert scale, ranging from "strongly disagree" to "strongly agree," all measures were evaluated.

The developed model consists of two reflective constructs, namely BPC and OR. The dynamic AIS capability as a second-order formative construct entails three dimensions: (1) flexible AIS, (2) complementary BI system, and (3) AIS-related human resource competency.

According to previous research by Uwizeyemungu and Raymond [99] and Prasad and Green [9], we used six measures to measure the flexibility of AIS. These measures requested respondents to assess the extent to which an AIS temporarily affected the organization's time to react to change, the extent variety to which an AIS affected the variety of responses available to the organization to cope with both anticipated and unforeseen changes, and the extent to which hardware, software, and data can be separated and combined to support the development of new AIS infrastructure systems. BI measurement was derived from the study by Prasad and Green [9] and Torres et al. [100]. AIS-related

human resource competency was based on the existing measurement using a five-item scale adapted from [9,101].

This research looks at a specific set of organizational capabilities known as business process capabilities. These capabilities characterize an organization's ability to create market value in distinctive ways by utilizing resources [50] and by analyzing internal and external resources. This spanning capability enables businesses to capitalize on important assets, prevent potential flaws, seize market opportunities, and counteract external threats. BPC is a first-order concept that underpins an organization's activities and signals the beginning of purposeful learning. BPC was assessed on BP outside-in capability, BP inside-out capability, and BP spanning capability with twelve question items.

There are two promising techniques to improving our understanding of OR: (1) process methods, which highlight the dynamic nature of resilience by defining multiple phases of resilience, and (2) resilience capacity studies, which provide an overview of resilience's internal functioning [40]. We believe that combining these two approaches will result in a more comprehensive understanding of the phenomena of resilience as well as a solid foundation for empirical study on the creation of resilient organizations. As a result, OR is founded on the concept of "resilience as a process" and the idea of resilience as the only combination of organizational capacity and routine. Based on this, we refer to the three phases of resilience, namely anticipation, adaptation, and recovery, as the measurement for OR.

## 5. Results

### 5.1. Common Method Bias

To ensure that the collected data do not contain common method bias (CMB), as recommended by Podsakoff et al. [102], In this study, Harman's single factor was used. Exploratory factor analysis was used to examine 42 scale items provided in the final measurement model, which were then measured using an unrotated factor solution. The single factor is not present, as evidenced by the fact that the first factor accounted for 41.647 percent of the variance, which is less than the 50% threshold number [102]. Thus, there is no existence for the CMB in the collected data.

### 5.2. Assessment of the Reflective Measurement Model

We used PLS-SEM to evaluate the structural and measurement model. The assessment of the measurement model was undertaken by evaluating the reliability and validity (including convergent and discriminant validity) of the collected data. The reliability was assessed through Cronbach's alpha and composite reliability (CR) [103]. As per the readings in Table 2, no issues were raised in terms of Cronbach's alpha and CR, as all the values were above the threshold value of 0.7 [104]. For the convergent validity, it is evident from Table 2 and Figure 1 that the values of factor loading were all above the threshold value of 0.6 [104]. However, during measurement model evaluation, BPC9 factor loading was low and affected the BPC's AVE. According to Hair et al. [103] instructions, any factor loading that is below 0.60 and affects the construct's AVE should be eliminated from the construct. Thus, removing BPC9 increased the BPC's AVE. In addition, the values of the average variance extracted (AVE) were all above the threshold value of 0.5 [103]. Similar threshold values were adopted in previous studies on information systems [25,105–112]. Therefore, the convergent validity was confirmed.

Concerning the discriminant validity, this research computes the Heterotrait–Monotrait Ratio (HTMT) as an alternative technique, as recommended by [113]. The results in Table 3 show that all the values were less than the threshold value of 0.85 [113], and thus, there were no issues regarding the discriminant validity.

**Table 2.** Reflective Measurement Model Assessment.

| Constructs | Items | Loadings | CA | CR | AVE |
|---|---|---|---|---|---|
| Fixable AIS | AISF1 | 0.871 | 0.899 | 0.923 | 0.667 |
| | AISF2 | 0.795 | | | |
| | AISF3 | 0.842 | | | |
| | AISF4 | 0.836 | | | |
| | AISF5 | 0.845 | | | |
| | AISF6 | 0.697 | | | |
| AIS-related Human Competency | AISRH1 | 0.838 | 0.877 | 0.910 | 0.670 |
| | AISRH2 | 0.837 | | | |
| | AISRH3 | 0.767 | | | |
| | AISRH4 | 0.833 | | | |
| | AISRH5 | 0.816 | | | |
| BP Capabilities | BPC1 | 0.667 | 0.926 | 0.937 | 0.579 |
| | BPC10 | 0.815 | | | |
| | BPC11 | 0.843 | | | |
| | BPC12 | 0.820 | | | |
| | BPC2 | 0.704 | | | |
| | BPC3 | 0.825 | | | |
| | BPC4 | 0.648 | | | |
| | BPC5 | 0.797 | | | |
| | BPC6 | 0.639 | | | |
| | BPC7 | 0.754 | | | |
| | BPC8 | 0.819 | | | |
| Complementary BI System | CBI1 | 0.855 | 0.865 | 0.918 | 0.788 |
| | CBI2 | 0.899 | | | |
| | CBI3 | 0.908 | | | |
| Organizational Resilience | OR1 | 0.739 | 0.941 | 0.948 | 0.532 |
| | OR10 | 0.750 | | | |
| | OR11 | 0.747 | | | |
| | OR12 | 0.703 | | | |
| | OR13 | 0.719 | | | |
| | OR14 | 0.645 | | | |
| | OR15 | 0.739 | | | |
| | OR16 | 0.722 | | | |
| | OR2 | 0.761 | | | |
| | OR3 | 0.769 | | | |
| | OR4 | 0.667 | | | |
| | OR5 | 0.730 | | | |
| | OR6 | 0.736 | | | |
| | OR7 | 0.770 | | | |
| | OR8 | 0.725 | | | |
| | OR9 | 0.737 | | | |

**Table 3.** Heterotrait–Monotrait Ratio (HTMT) Results.

| | AIS-Related Human Competency | BP Capabilities | Complementary BI System | Fixable AIS | Organizational Resilience |
|---|---|---|---|---|---|
| AIS-related Human Competency | | | | | |
| BP Capabilities | 0.523 | | | | |
| Complementary BI System | 0.715 | 0.518 | | | |
| Fixable AIS | 0.796 | 0.599 | 0.697 | | |
| Organizational Resilience | 0.772 | 0.729 | 0.767 | 0.798 | |

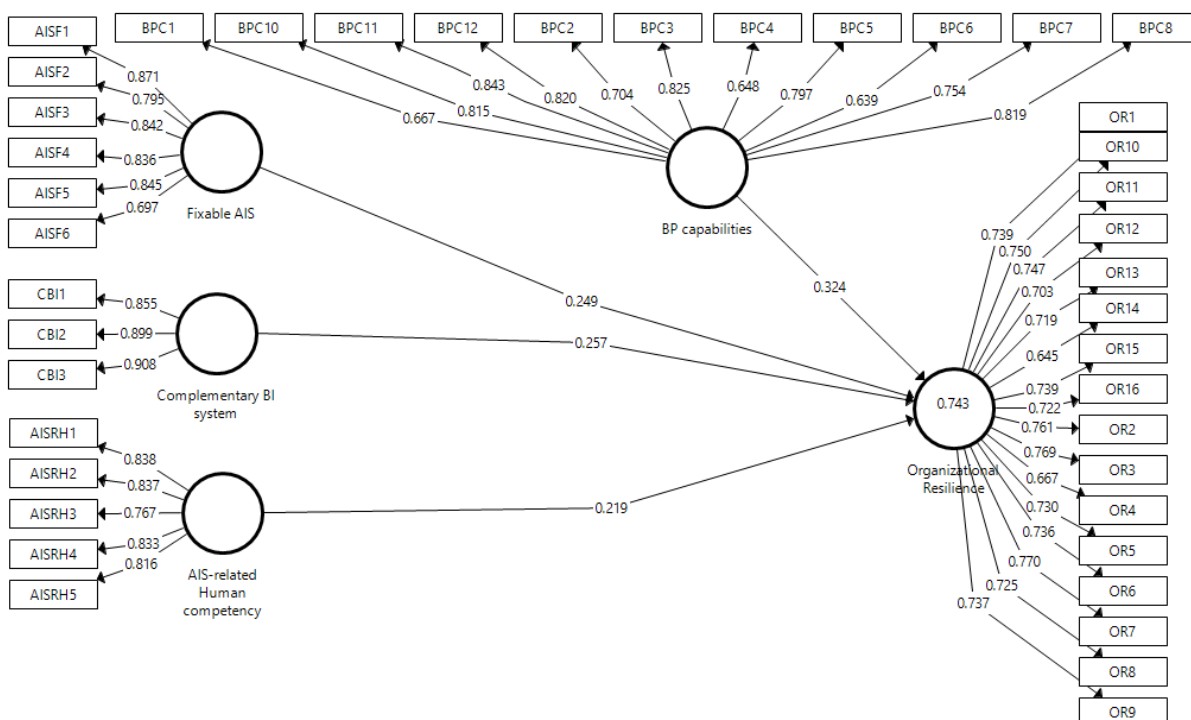

**Figure 1.** Measurement Model.

### 5.3. Assessment of Dynamic AIS Capability as a Formative Second-Order Construct

In the current study, dynamic AIS capability was measured as a second-order formative construct with three first-order reflective constructs. The three first-order constructs are AIS-related human competency, complementary BI system, and flexible AIS measured by reflective indicators. The purpose of this study is to investigate the reliability and validity of the measurement model in the second order, as suggested by Chin [114]. As recommended by Ringle et al. [115], to create the second-order construct, a two-stage strategy combining the repeated indicator approach and the use of latent variable scores (LVS) should be used to examine the measurement model of the model that includes formative components. Endogenous formative structures that are part of the structural model as well as higher-order constructs require a two-stage method (HOC) [116].

The variance inflation factor (VIF) was evaluated to guarantee that there were no collinearity difficulties in the f dynamic AIS capabilities formative construct. The variance inflation factor value should be below 5 [117], and the tolerance value should be higher than 0.20 (Hair et al., 2016 [104]).

The collinearity assessment of the formative second-order construct is shown in Table 4. Collinearity must be assessed to ensure that the constructs are not measuring the same belief variables. The variance inflation factor (VIF) values for each formative construct are lower than the threshold value of 3.3, as indicated in Table 4 [116].

**Table 4.** Collinearity Assessment for Second-order Construct.

| Path | VIF |
|---|---|
| AIS-related Human Competency | 2.289 |
| Complementary BI system | 1.814 |
| Flexible AIS | 2.258 |

Following that, the importance of each formative construct's weight in explaining the first-order constructs was evaluated. Table 5 shows the bootstrapping findings for each of the formative second-order structures using sub samples of 5000 cases, with weights and path coefficients for each [103,116]. Table 5 shows the estimation of the AIS capability

as a second-order construct. The weights from the second-order construct (dynamic AIS capability) to the three first-order factors (AIS-related human competency, complementary BI system, and flexible AIS) are significant (0.286, 0.344, and 0.506), with a very high T ratio of 2.713, 4.046, and 4.849 for AIS-related human competency, complementary BI system, and flexible AIS, respectively. The results suggest that the second-order construct adequately captures the link between first-order components [114].

**Table 5.** The Weights and Path Coefficients for Each of The Formative Second-order Constructs.

| Relationship | Path Coefficients | T Statistics | *p*-Value |
|---|---|---|---|
| AIS-related Human Competency -> Dynamic AIS Capability | 0.286 | 2.713 | 0.003 |
| Complementary BI System -> Dynamic AIS Capability | 0.344 | 4.046 | 0.000 |
| Flexible AIS -> Dynamic AIS Capability | 0.506 | 4.849 | 0.000 |

*5.4. Assessment of the Structural Model Hypotheses Testing*

The structural model was evaluated using coefficients of determination ($R^2$), *t*-values, $f^2$ effect sizes, predictive relevance ($Q^2$), and path coefficient significance [103,116].

The $R^2$ number indicates how well the independent components are able to predict or explain the dependent constructs. The higher the $R^2$, the greater the model's predictive power. Table 6 shows that the $R^2$ of organizational resilience is 0.736. This indicates that dynamic AIS capability, AIS-related human competency, complementary BI system, and fixable AIS accounted for 73.6% of the variance of the organizational resilience. Furthermore, the effect size or f-square can be used to determine the strength of an independent construct's effect on the dependent construct in a model ($f^2$): $f^2 = 0.02$, which is regarded as a little influence; $f^2 = 0.15$, which is categorized as a medium effect; and $f^2 = 0.35$, which is classified as a significant effect, according to Hair Jr. et al. [116]. The results indicate that dynamic AIS capability has a large influence on organizational resilience, with a size effect of 1.037. Meanwhile, AIS-related human competency, complementary BI system, and fixable AIS have a small effect on the organizational resilience, with values of 0.081, 0.138, and 0.098, respectively. In addition, the $Q^2$ value should be considered as an extra metric for determining the magnitude of $R^2$ values. For some reflecting endogenous latent variables, $Q^2$ values greater than zero suggest the route model's predictive importance for this construct [103,116]. For each endogenous construct, a blindfolding process was employed to produce cross-validated redundancy measures. The $Q^2$ predictive value of organizational resilience is 0.376, exceeding zero and thus indicating that the model has predictive relevance for the organizational resilience construct.

**Table 6.** Structural model: Path coefficients and *t*-statistics analysis.

| H | Path | Path Coefficient | *t*-Value | *p*-Value | $f^2$ | $R^2$ | $Q^2$ |
|---|---|---|---|---|---|---|---|
| H1 | Dynamic AIS Capability -> Organizational Resilience | 0.632 | 8.094 | 0.000 | 1.037 | | |
| H1.1 | AIS-related Human Competency -> Organizational Resilience | 0.219 | 3.470 | 0.000 | 0.081 | 0.736 | 0.376 |
| H1.2 | Complementary BI System -> Organizational Resilience | 0.257 | 3.825 | 0.000 | 0.138 | | |
| H1.2 | Fixable AIS -> Organizational Resilience | 0.249 | 3.714 | 0.000 | 0.098 | | |

The hypotheses were further confirmed by performing significance tests on path coefficients. The precision of the PLS estimates was estimated using a bootstrapping analysis. The results of path coefficients in Table 6 show that dynamic AIS capability, AIS-related human competency, complementary BI system, and fixable AIS have a significant influence on the organizational resilience.

*5.5. Mediation Analysis*

In order to evaluate the mediation effect in the proposed model, based on the recommendation of Preacher and Hayes [118], bootstrap analysis method was employed as a more recent and statistically robust approach [116]. For the indirect effect on the relationship between dynamic AIS capability and organizational resilience for the mediator effect of BPC and CMC, a bootstrapping procedure with 5000 sub-samples was performed. The mediation test results demonstrated that hypothesis H4 was supported: (1) the indirect effect is significant ($\beta = 0.186$, $t = 3.008$, $p < 0.001$), and (2) neither of the 95% bootstrapped confidence intervals includes zero. Therefore, the findings concluded that there is a mediation effect of BP capabilities between dynamic AIS capability and organizational resilience.

## 6. Discussions and Conclusions

As regards our empirical results on the impact of dynamic AIS capability on organizational resilience, the business process is consistent with previous studies, but they also offer new findings on the association between dynamic AIS and business process capabilities to improve organizational resilience (such as the relationship between AIS-human resources competency and BPC). Consistent with the RBT and DCV perspective, the findings highlighted the critical role of dynamic AIS capability in improving organizational resilience. The results strongly support the claim that an organization's dynamic AIS capabilities—both flexible AIS, complementary BI system, and AIS-related human resource competency—can help the organization improve its resilience. This finding is in line with previous research that has suggested that IT capabilities might help a business increase its resilience or agility (see [94,119,120].

We identified capabilities that serve as a mediator between an organization's dynamic AIS capability and its business process capabilities. These capabilities include outside-in, inside-out, and spanning capabilities. We discussed the impact of dynamic AIS capability of each of these capabilities (i.e., business process capabilities) for enhancing organizational resilience. The findings of the study show how dynamic AIS capabilities can help develop business process capabilities. This finding is consistent with the literature in the realms of resources-based theory and dynamic capabilities as well as previous studies (see [94,120–122]).

The findings of the mediating effect test indicate partial mediation of BPC on the relationship between dynamic AIS capability and organizational resilience. This finding strengthened the idea of a ranking of capabilities, which suggests that lower-level capabilities can help an organization in developing higher-order ones [123]. This finding strengthened the idea of a ranking of capabilities, which suggests that lower-level capabilities can help an organization in developing higher-order ones. This view is consistent with previous studies, which have stated that the impacts of IT capabilities (i.e., dynamic AIS capability), such as lower-order capabilities, on organizational resilience are mediated by business processes capabilities, such as higher-order capabilities (see [124,125]).

This study is based on the notion that organizations can rely on dynamic capabilities to remain competitive when facing complex and uncertain business environments to remain competitive by being adaptive. The study also employs RBT, which proposes that dynamic AIS capability is a "resource capability" with tremendous implications for BPC and CMC as well as organizational resilience [45]. Our results support the notion that IT resources alone will not have an influence on information or result in a positive improvement in performance unless IT is integrated into the business processes' roles [126]. Organizations with improved BPC can easily anticipate market demands, identify external competitors, establish long-term relationships with external stakeholders, and respond quickly to market changes. It also helps organizations sustain competitiveness and use their valuable strengths to identify threats and exploit market opportunities.

Following Mikalef and Pateli [127], our study suggests that AIS is not only a resource capacity but also a dynamic capability and a precursor to the business processes of an

organization and organizational resilience. As part of business processes, the challenge for organizations is to determine not only how to acquire AIS infrastructure but also how to consider external and internal environments when building dynamic AIS capability. In addition, there is a need for a swift update of the resources that establish the dynamic capability, and the organization of the resources must be in a way that responds rapidly to threats and opportunities. The mediating role of BPC on the indirect relationship between dynamic AIS capability and organizational resilience was also evident. Another indication of the research findings is that the understanding of the mediating role of CMC is crucial towards understanding the impact of dynamic AIS on organizations. AIS exercises its influence on the organization through complementary relations with other assets and capabilities of the organization [52]. Despite the impact an AIS system can have on an organization, the focus of many scholars has been on organizational financial performance, but this study observed the positive role of dynamic AIS systems on organizational capabilities and organizational resilience.

The outcome of this study suggests the need for a best-fit model of reality that can suit the dynamic accounting environment. Contrary to previous studies that mainly base their conceptuality of IT capabilities on RBT, the creation of dynamic IT-enabled capabilities is more appropriate to describe how integrated IT in organizational business processes can contribute to achieving and maintaining a competitive advantage.

### 6.1. Theoretical Implications

In general, the development of a model to understand the impact of dynamic AIS capability on organizational resilience is a significant theoretical contribution. This should be a beneficial initial step in furthering our understanding of the resilience capability building process and its performance consequences through the development of dynamic AIS capability based on the RBT and DCV theories. This work makes significant contributions to the field of AIS in particular. More specifically, this work incorporates the RBT and DC theories to give a theoretical foundation for assessing the impact of dynamic AIS capabilities on organizations. Both theories have been discussed comprehensively in ISs management and operations literature; however, they are yet to be detailed on dynamic AIS systems. This study relied on both theories to develop a theoretical framework to evaluate the impact of dynamic AIS capability on BPC and OR.

Second, this research directs attention toward building dynamic AIS capability by the synergy between AIS and related competencies. Previous researchers have focused on studying generic AIS alone without considering that IS remains a part of an organization's dynamic resources rather than standalone resources. Third, another implication of this work is on the dynamic capability aspect of the RBV. This study articulates the role of IT capabilities in organizational resilience and processes capabilities development (Fawcett et al. 2011). This study contributes in two ways to the DCV literature; first, in line with van de Wetering et al. (2017) [3] and Liu et al. (2013) [87], this study found AIS as both a resource capability and a dynamic capability; it is also a prerequisite to organizational resilience and business processes capabilities of an organization. Secondly, this study relied on the DCV to posit that agility or resilience is a dynamic capability that is enabled by numerous other capabilities [128].

Finally, the study bridges different studies on dynamic AIS capability, BPC, and organizational resilience. First, it empirically tests their relationships in the resource-based theory and dynamic capabilities context. Second, existing research has looked at the direct and indirect consequences of dynamic capability on organization performance [129,130]. However, limited studies have simultaneously examined dynamic AIS capability and these two types of effects. Hence, this study lends empirical support to the notion of OR being directly or indirectly influenced by dynamic capability.

### 6.2. Managerial Implications

The empirical results of this study appear to have important implications on individuals (such as managers, accountants) and organizations. First, the findings in this study suggest that organizations can benefit from building a dynamic AIS environment in the following ways. Accounting is indeed considered a support function; however, it may be influenced by a changing business environment. Organizations need to exploit modern IT tools, such as e-Service and web service, mobile devices, cloud computing, environmental scanning, BI, enterprise application integration, BP management, big data, and other tools, to update their business processes and AIS, as these accounting processes require ongoing evaluation. There is a need for a more committed and persistent approach to the control and management of AIS. Again, the concept of complementary assets investment (such as business processes models, decision-making process, and training) is more pertinent in an AIS environment, and AIS staff resources are of special significance in this regard. Hence, organizations must be more liberal during the recruitment and training of AIS staff to ensure they can contribute to the maintenance of the AIS's fit to the organization's demands.

Second, although dynamic AIS capability (flexible AIS, complementary BI, and AIS-related human resource competency) is a fundamental element of final organization performance, it is expected that organizations should improve their BPs for a prolonged period; otherwise, there may be no improvement in organizational performance regardless of the level of capital investments [131].

Third, managers also need to devise how to exploit various types of dynamic AIS capabilities in an efficient and effective manner to foster OR. Our results showed significant positive effects of complementary BI system and human IT capability on OR, whereas flexible AIS is low in the presence of high organizational capability to integrate IT, business processes, and strategy [132]. These findings can help managers integrate dynamic AISs into their business routines to improve operational capabilities and organizational resilience as well as decipher how to manage a hierarchy of capabilities to achieve excellence in operations.

Fourth, the confirmation that dynamic AIS and business process skills are complimentary in establishing organizational resilience should serve as a reminder to managers that they need to improve their AIS infrastructure's integrative capabilities. Fifth, the results of the study provide managers and policymakers with a practical awareness of the fact that the most crucial resources within an organization for building dynamic AIS are flexible, and the combination of AIS and AIS-related competencies of personnel as well as the use of AIS and BI systems allows a business to become more robust.

Finally, managers in emerging economies such as Malaysia are urged to deploy IT-enabled resources by investigating the context of IT use. In order to survive in today's dynamic business environment, companies must spend in improving their IT capabilities. Most large companies put a great deal of money into capability development initiatives and make good use of IT resources in their business processes. As a result, this research has practical implications for managers in underdeveloped nations who want to improve their operational capabilities and attain OR by incorporating IT resources into their daily operations.

### 6.3. Limitations and Avenues for Future Research

First, the results of this study are in some ways based on cross-sectional data. Since IT-enabled OR development is a gradual process, there is no historical information on the impact of the independent variables on the dependent variable over time. With this cross-sectional design approach, the results can be easily analyzed with care, as there are no chances of causality being inferred from cross-sectional data [133]. This cross-sectional study can be used as a foundation for future longitudinal studies to replicate and compare the methods used in this work in different situations (e.g., developed countries). A longitudinal study can be employed to address the study's cross-sectional nature by determining changes before and after the adoption of dynamic AIS systems.

Second, although each component of dynamic AIS capability effectively enhances OR, we measured the dynamic AIS capability using an indirect approach; this aimed to identify the presence of dynamic AIS capability and its relationship with BPC and CMC in order to highlight the significance of this construct. Meanwhile, only three critical dynamic AIS capabilities, namely BI system, flexible AIS, and AIS-related human resource competence, were discussed, as the other dimensions in the broad spectrum of dynamic AIS capability were not discussed. A direct measure of dynamic AIS capabilities will provide opportunities for future research. Meanwhile, future research can also investigate other IT or organizational capabilities that can influence OR.

Finally, while we know much about how organizations cope with unexpected events, we know less about the resilience stages and how dynamic AIS as a source of knowledge may impact them. Therefore, management scholars should focus more on other research fields and incorporate the existing ideas and findings in their studies. Studies on the resilience capabilities in organization and management studies are still lacking [40]. The essential role of organizational knowledge (such as dynamic AISs), structure, and culture in organizational capability development towards dealing with unexpected and threatening situations is yet to be acknowledged. To close the OR research gap, future studies can focus on the less-explored aspects of the resilience process, such as clarifying the essential role of organizational knowledge, culture, and structure in organizational capability development. Future studies may also investigate the level of preparation of organizations for unexpected events, how they accept problems, and how they learn from such problems using DAIS.

**Author Contributions:** Formal analysis, M.A.A.-S.; Supervision, R.A. and K.A.A.; Writing—original draft, A.S.A.-M.; Writing—review & editing, M.A.A.-S. All authors have read and agreed to the published version of the manuscript.

**Funding:** This research was funded by Universiti Kebangsaan Malaysia, grant number EP-2018-001.

**Institutional Review Board Statement:** Not applicable.

**Informed Consent Statement:** Not applicable.

**Acknowledgments:** Authors are grateful for financial support from the Universiti Kebangsaan Malaysia (UKM) for supporting this research and providing research facilities.

**Conflicts of Interest:** The authors declare no conflict of interest.

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
