# Peer review of "The Impact of Dynamic Accounting Information System on Organizational Resilience: The Mediating Role of Business Processes Capabilities"

_sustainability, doi:10.3390/su14094967_

Round 1
Reviewer 1 Report
Dear Author(s),
Please find below my concerns and recommendations regarding your manuscript proposal entitled "The Impact of Dynamic Accounting Information System on Organizational Resilience: The Mediating Role of Business Processes Capabilities".
1. The Introduction section is pretty-well written, but you should better highlight the following important aspects:
- the research gap: based on the previous results from the literature, please clearly define and describe the research gap covered by your article, so that the readers understand what your contribution will be. At this moment, at rows 91-93 you have a try, but you need to describe it more clearly ("This study also looks into how dynamic AIS capability affects OR by identifying a research gap in AIS business value and its impact on OR, as mediated by BPC.")
- the research question(s): please describe and define the research question(s) because every scientific article must respond to a research question.
- the research goal: define the specific goal of your research.
2. The font size of the rows 381-383 and 391-393 seems to be different from the rest of the text. The same remarks for the rows 593-637.
If it's possible, please revise this editing issue.
3. I recommend you to insert a new table at the end of the section "4.2. Measurement items" or at the end of the article (Appendix) where you describe every item (from AISF1 to OR16), so that the readers know them.
4. Rows 562-564 you say: "As shown in Table 7, the mediation test results demonstrated that hypothesis H4 was supported; (1) the indirect effect is significant (β=0.065, t=2.347, p<0.001) and (2) neither of the 95% bootstrapped confidence intervals includes zero."
Please take a look at the table 7 and compare the values from the tables to the values from the text. They are different (Beta, t-value). Please revise this issue.
5. I recommend you to include in your manuscript the following valuable references: https://doi.org/10.3390/su132212706, https://doi.org/10.3390/admsci10040096, https://doi.org/10.4018/JGIM.294905, http://www.transformations.knf.vu.lt/47a/article/amod, https://www.iarj.in/index.php/ijracm/article/view/34, https://doi.org/10.1080/12460125.2020.1861772, https://doi.org/10.17512/pjms.2020.21.2.03.
By quoting these references, you will broaden the general context of your research.
6. In the chapter "5.2. Assessment of the reflective measurement model", between rows 476-478, you say:
"As per the readings in Table 3, no issues were raised in terms of Cronbach’s alpha and CR, as all the values were above the threshold value of 0.7 [93]."
I think that the values are in "Table 2. Reflective Measurement Model Assessment.", not in table 3.
Please revise and synchronize the numbers of the tables.
7. At rows 488-491 you say: "Concerning the discriminant validity, this research computes the Heterotrait-Monotrait Ratio (HTMT) as an alternative technique as recommended by [94]. The results in Table 4 showed that all the values were less than the threshold value of 0.85 [94], and thus, there were no issues regarding the discriminant validity."
The correct table is "Table 3. Heterotrait-Monotrait Ratio (HTMT) Results", not Table 4.
Please revise and correct the number of the table.
Dear Author(s),
Please consider all the above remarks as being constructive recommendations in order to improve the general quality of your manuscript proposal.
Kind Regards!
Author Response
We thank you for the time in reviewing the manuscript entitled “The Impact of Dynamic Accounting Information System on Organizational Resilience: The Mediating Role of Business Processes Capabilities”. Also, for the detailed and insightful comments, allow us to substantially develop our manuscript from the initial manuscript. We have carefully reviewed the comments and have revised the manuscript for the second round. We have highlighted the changes within the manuscript. Here is a point-by-point response to the reviewers’ comments and concerns.
Thank you and we highly appreciate your kind consideration and cooperation to approve and list our manuscript for publication shortly.
Yours Sincerely,
Reviewer 1
Comments and Suggestions for Authors
Dear Author(s),
Please find below my concerns and recommendations regarding your manuscript proposal entitled "The Impact of Dynamic Accounting Information System on Organizational Resilience: The Mediating Role of Business Processes Capabilities".
- The Introduction section is pretty well written, but you should better highlight the following important aspects: - the research gap: based on the previous results from the literature, please clearly define and describe the research gap covered by your article, so that the readers understand what your contribution will be. At this moment, at rows 91-93 you have a try, but you need to describe it more clearly ("This study also looks into how dynamic AIS capability affects OR by identifying a research gap in AIS business value and its impact on OR, as mediated by BPC.") the research question(s): please describe and define the research question(s) because every scientific article must respond to a research question. - the research goal: define the specific goal of your research.
Response
Thank you for the positive feedback. The research gap has been modified and highlighted in the revised version of the manuscript.
- The font size of the rows 381-383 and 391-393 seems to be different from the rest of the text. The same remarks for the rows 593-637.
If it's possible, please revise this editing issue.
Response
Thank you for highlighting this point. The paper format has been updated based on the sustainability instructions for authors.
- Rows 562-564 you say: "As shown in Table 7, the mediation test results demonstrated that hypothesis H4 was supported; (1) the indirect effect is significant (β=0.065, t=2.347, p<0.001) and (2) neither of the 95% bootstrapped confidence intervals includes zero."
Please take a look at table 7 and compare the values from the tables to the values from the text. They are different (Beta, t-value). Please revise this issue.
Response
We agree with the reviewer concerning this comment, the text on the revised version of the manuscript has been updated based. Also, we converted Table 7 into a narrative description (It makes no sense to keep a table with only one row) to address the comment of reviewer 2.
- I recommend you to include in your manuscript the following valuable references: https://doi.org/10.3390/su132212706, https://doi.org/10.3390/admsci10040096, https://doi.org/10.4018/JGIM.294905, http://www.transformations.knf.vu.lt/47a/article/amod, https://www.iarj.in/index.php/ijracm/article/view/34, https://doi.org/10.1080/12460125.2020.1861772, https://doi.org/10.17512/pjms.2020.21.2.03.
By quoting these references, you will broaden the general context of your research.
Response
Thanks for the suggestion. All suggested references have been added to the revised version of the manuscript.
- In the chapter "5.2. Assessment of the reflective measurement model", between rows 476-478, you say: "As per the readings in Table 3, no issues were raised in terms of Cronbach’s alpha and CR, as all the values were above the threshold value of 0.7 [93]."
I think that the values are in "Table 2. Reflective Measurement Model Assessment.", not in table 3. Please revise and synchronize the numbers of the tables.
Response
Thank you for raising this point. The table numbers have been corrected.
- At rows 488-491 you say: "Concerning the discriminant validity, this research computes the Heterotrait-Monotrait Ratio (HTMT) as an alternative technique as recommended by [94]. The results in Table 4 showed that all the values were less than the threshold value of 0.85 [94], and thus, there were no issues regarding the discriminant validity."
The correct table is "Table 3. Heterotrait-Monotrait Ratio (HTMT) Results", not Table 4.
Please revise and correct the number of the table.
Response
Thank you for highlighting this. The typo errors have been corrected.
Dear Author(s),
Please consider all the above remarks as being constructive recommendations in order to improve the general quality of your manuscript proposal.
Response
Thank you once again for your insightful comments that allow us to substantially develop our manuscript.
Reviewer 2 Report
Title
The Impact of Dynamic Accounting Information System on Organizational Resilience: The Mediating Role of Business Processes Capabilities
Overall comments
In this work, the authors study the role of business process capabilities in modulating the link between dynamic Accounting Information Systems (AIS) capability and organizational resilience, based on 144 matched questionnaires selected from large companies from various sectors listed on the Bursa Malaysia. Based on their results, the authors claims that the impact on organizational resilience is positively affected by mediated of business process capabilities.
The study also suggest that dynamic AIS capability has an impact on organizational resilience. They also claim the existence of a link between dynamic AIS and business process capacity to improve organizational resilience. Moreover, they also claim, based on the achieved results, that an organization's dynamic AIS capabilities can help the organization to improve its resilience.
The work, ends up with the presentation and discussion of some limitations, founded here.
Overall speaking, the paper seems to be well-organized, containing all the expected components, namely the Introduction, Hypothesis, Research Methods, Discussion and Analysis of results, and Conclusions.
Despite the author’s results seems to me convincing, given the purpose of the work, I’m still reluctant about the relation between the scope of this paper and the scope of the journal, although the existence of the research topic of organizational resilience.
After all, what is the main contribution of this paper in terms of sustainability or sustainable development?
However, and in general, the authors have answered to the research question stated here, despite the absence of some control experiments in methodological section. The authors only present more details regarding the methods used on section 5 (results), instead of presenting on section 4 (methodology).
However, the relevance of the subject treated here (organization resilience) is also high in the present.
Some recommendations regarding this issue, can be found it below.
Some recommendations of improvement:
Strong points:
- The relevance of the subject
- Case study and data used.
- Future work
- Literature review: Although it could be improved, since that some of the literature used here are old (2017 or less), which support the need of update, in order to reinforce the importance of this study
- Discussion of results
Weak points:
- Conclusions – Although the quality’s discussion within the obtained results, seems to me reasonable, the conclusions section, could be improved to reinforce the obtained answers/paper’s outcomes, regarding the main question initially pointed on the introduction.
- Research methods used – Please see the comments mentioned above on “overall comments”.
- The relationship with sustainability or sustainable development – should be better discussed this part, in order to highlight the paper’s main contribution, regarding the role of the organization’s resilience on the organization’s sustainability.
Additional comments
Please, carefully revise the entire text. There are some orthographic errors over the entire text.
Some examples:
Line 25 – (..)”capacity to improve organisational resilience. “(..)
Line 27 – (..)”help the organization to improve its resilience” (..) instead of (..)”help the organisation improve its resilience” (..)
Please, carefully revise the entire text. Some acronyms aren’t previously defined (e.g. line 27à(..)”complementary BI system” (..)
Author Response
We thank you for the time in reviewing the manuscript entitled “The Impact of Dynamic Accounting Information System on Organizational Resilience: The Mediating Role of Business Processes Capabilities”. Also, for the detailed and insightful comments, allow us to substantially develop our manuscript from the initial manuscript. We have carefully reviewed the comments and have revised the manuscript for the second round. We have highlighted the changes within the manuscript. Here is a point-by-point response to the reviewers’ comments and concerns.
Thank you and we highly appreciate your kind consideration and cooperation to approve and list our manuscript for publication shortly.

Reviewer 3 Report
This Research presents a very interesting analysis of the Impact of Dynamic AISs on Organisational Resilience.
The article is well structured and it includes relevant conclusions, limitations along with interesting theoretical and practical implications.
Relevant references are included and the literature is supporting the analyses.
My only (minor) concerns are listed below:
- A short definition of Dynamic AIS is missing in the introduction (this could better orientate readers who do not know the topic) before extensively explain it in the literature
- typing mistake in line 97 (aa)
- I would have expected the research to make more extensive reference to the agile philosophy (since it is mentioned twice) that could positively affect the systems
- I would rather turn Table 7 into a narrative description (It makes no sense to keep a table with only one row)
Author Response
Thank you for giving us the opportunity to submit a revised draft of the manuscript “The Impact of Dynamic Accounting Information System on Organizational Resilience: The Mediating Role of Business Processes Capabilities” for publication in the Sustainability. We appreciate the time and effort that you dedicated to providing feedback on our manuscript and are grateful for the insightful comments on and valuable improvements to our paper. We have highlighted the changes within the manuscript. Here is a point-by-point response to the reviewers’ comments and concerns.
Thank you and we highly appreciate your kind consideration and cooperation to approve and list our manuscript for publication shortly. Please see the attachment for details of the revised manuscript.
Comments and Suggestions for Authors
This Research presents a very interesting analysis of the Impact of Dynamic AISs on Organisational Resilience. The article is well structured and it includes relevant conclusions, limitations along with interesting theoretical and practical implications. Relevant references are included and the literature is supporting the analyses.
Thank you for the positive feedback.
My only (minor) concerns are listed below:
- A short definition of Dynamic AIS is missing in the introduction (this could better orientate readers who do not know the topic) before extensively explain it in the literature.
Thank you to highlight this, we do agree with the reviewer, to add the definition of Dynamic AIS to the introduction section.
- typing mistake in line 97 (aa)
Thank you for highlighting this. The typo errors have been corrected.
- I would have expected the research to make more extensive reference to the agile philosophy (since it is mentioned twice) that could positively affect the systems.
Thanks for the suggestion. More recent references have been added to the revised version of the manuscripts.
- I would rather turn Table 7 into a narrative description (It makes no sense to keep a table with only one row).
Thanks for the suggestion. Table 7 has been changed into a narrative description as you suggested.
Round 2
Reviewer 1 Report
Dear Author(s),
I have read the revised version of the manuscript and I consider that you addressed all my constructive recommendations from the previous round of review.
Now I have only one minor remark: in the Conclusions sections, the font between rows 608-652 seems to be different from the rest of the document. Please revise and correct this minor issue.
Kind Regards!
Reviewer 2 Report
In general, the authors have made significant efforts to improve the quality of the manuscript, by carefully revising all the text and by adding new issues, not only to reinforce the paper's main contribution, but also, by improving the discussion section, considering my initially comments.